# A First Approach to Determine if It Is Possible to Delineate In-Season N Fertilization Maps for Wheat Using NDVI Derived from Sentinel-2

**Asier Uribeetxebarria \*, Ander Castellón and Ana Aizpurua**

NEIKER-Basque Institute for Agricultural Research and Development, Berreaga 1, 48160 Derio, Biscay, Spain; acastellon@neiker.eus (A.C.); aaizpurua@neiker.eus (A.A.)

\* Correspondence: auribeetxebarria@neiker.eus

**Abstract:** Adjusting nitrogen fertilization to the nutritional requirements of crops is one of the major challenges of modern agriculture. The amount of N needed is mainly determined by crop yield, so yield maps can be used to optimize N fertilization. As the adoption of yield monitors is low among farmers, implementation of this approach is still low. However, as the Normalized Difference Vegetation Index (NDVI) is related to grain yield, the main objective of this work was to identify at which wheat growth stage a moderate agreement between NDVI and yield is obtained. For this, NDVI images obtained from Sentinel-2 were used, and the evolution of concordance was analyzed in 13 classified parcels of wheat employing the Kappa index (KI). In one-third of the plots, a moderate agreement (KI > 0.4) was reached before the stem elongation growth phase (when the last N application was made). In another one-third, moderate agreement was reached later, in more advanced development stages. For the cases in which this agreement did not exist, an attempt was made to find the causes. The MANOVA and subsequent descriptive discriminant analysis (DDA) showed that the NDVI dates that contribute the most to the differentiation between plots with and without agreement between grain yield maps and NDVI images were those corresponding to tillering. The sum of the NDVI values of the tillering phase was significantly lower in the group of plots that did not show concordance. Sentinel-2 imagery was successful on 66% of plots for delineation of management zones after GS 30, and thus is useful for producing fertilization maps for the upcoming season. However, to produce in-season fertilization maps, further studies are needed to better understand the mechanisms that regulate the relation between yield and NDVI at early growth stages (<GS 30).

**Keywords:** tillering; Kappa Index; site-specific management zones; precision agriculture; fertilizer

## 1. Introduction

With an area cultivated of nearly 200 million hectares, wheat (*Triticum* sp.) is the second most widely grown crop in the world, [1] and wheat grain consumption accounts for 19% of the calories in the world's human diet. In Europe, using more productive varieties and N fertilizers, wheat yields have increased annually 1.5–2.5% for many decades, from an average of 2 t ha⁻¹ in 1900 to 7.5 t ha⁻¹ in 2000 [2]. During the last decades, N has been considered the most important nutrient to increase cereal production [3], in consequence, the use of nitrogen as a fertilizer has gradually increased. In 2002–2003, 142 million tons of nitrogen fertilizers were applied globally, and this amount increased to 175 million tons in 2009–2010. Furthermore, it is estimated to rise to 199 million tons by 2030 [4].

However, the excessive use of N fertilizers is related to environmental problems. On the one hand, they require a considerable amount of energy for their generation, and the N gases emitted after fertilization contribute to the greenhouse effect [5]. On the other hand, excessive application of N can provoke nitrate losses, which could pollute

groundwater and surface water bodies [6]. Considering the increasing need to supply food to a growing population and the potential environmental impact of N fertilizers, the optimal adjustment of N dosage is one of the challenges of modern agriculture. The optimum dose would be the quantity of fertilizer needed to obtain the maximum crop yield on each plot with the minimum risk for environmental damage.

Conventional agriculture manages fields homogeneously, without considering their spatial variation. However, the nutritional needs of the crop differ according to the area [7]. Generally, in more productive areas, wheat has higher N requirements, while nutritional needs are lower in areas where production is lower. Therefore, a good fertilizer adjustment strategy would have to consider crop variability and adjust the dose considering the potential production. Otherwise, productivity decreases finite resources such as water and fertilizers are misused, and detrimental impacts on the environment are produced [8].

Precision agriculture (PA) is a management strategy being more and more used [9] as it gathers, processes, and analyzes temporal, spatial and individual data, and combines them with other information to support management decisions according to estimated variability to improve resource use efficiency, productivity, quality, profitability, and sustainability of agricultural production [10]. As the PA takes into consideration the space to time variability of the crop, it can help to optimize N fertilization. For this purpose, it divides the plot into different site-specific management zones (SSMZ). Within each SSMZ, the properties of interest are similar, but different from those of other zones [11]. Once the plot is divided into SSMZ, a different fertilizer dose can be applied to each zone using variable rate management (VRA) [12]. Different SSMZ can be delineated for each agronomic practice (fertilization, pest management) in the same plot [13]. In this sense, Cordoba et al. [14] published a protocol to delineate multivariate homogeneous zones, where several levels of auxiliary information are considered.

Depending on the agricultural objective, different auxiliary information can be used to delineate the SSMZ. For example, apparent electrical conductivity [15], visible and near-infrared reflectance [16], and gamma-ray spectrometry [17] are used to measure soil variability. Thermal sensors can measure crop water state [18] or detect the stress created by pests [19]. Furthermore, sensors that provide information about the vegetative state of the crops are one of the most employed. Sensors are also widely used to obtain vegetation indices (VI). These indices are related to many crop properties, such as yield (biomass or grain yield). VI can be measured by active or passive sensors and can have different spatial and temporal resolution. For example, Maresma et al. [20] worked with multispectral imagery and VI to improve N management in maize fields.

In our edaphoclimatic conditions, the relationship between VI obtained from crop canopy active reflectance sensors, such as the RapidSCAN CS-45 (Holland Scientific) and wheat yield (grain t ha$^{-1}$) was analyzed by [21]. Being an active sensor and considering the short distance between the sensor and the crop, measurements are not affected by light conditions or the presence of clouds. Therefore, the most common use of this type of tool is the monitoring of experimental trials, in which the effects of different treatments are tested. As the measurements from these sensors are related to the information provided by satellites, their use makes it possible to project the knowledge gained in trials to commercial plots. However, these sensors are not the most suitable for measuring the variability of crop vigor over large areas or for adjusting fertilization.

A simple classification of passive sensors can be made according to their spatial resolution, starting with UAV-mounted cameras, which have the highest spatial resolution, continuing with aircraft-mounted sensors, and ending with satellite-mounted sensors, which can measure larger areas. A positive correlation with yield has been found for three sensor groups [22–24]. Since UAV images have a high resolution (±1 cm), they are suitable for scientific testing or monitoring small areas. However, a drawback is the need of powerful computers to analyze the data. Aerial images have an optimal resolution (25 cm) for work with medium-sized commercial areas. One disadvantage of these aerial images is

the need to contract commercial flights and their consequent economic cost that is not affordable for small farmers [25]. In addition, for satellite imagery to be usable in precision agriculture, Clevers et al. [26] suggested that the temporal resolution should not exceed two weeks while the spatial resolution should not exceed 20 × 20 m. Sentinel-2 is considered optimal for precision agriculture as the spatial resolution is 10 × 10 m and has a revisit period of 5 days [27]. However, the presence of clouds can be a disadvantage at certain latitudes, as constant coverage can interfere with crop monitoring [28].

The vegetation index time series are good indicators of vegetation growth and canopy behavior [29]. Time series derived from satellite images have been used for different purposes. For example, Potgieter et al. [30] used them to estimate crop area in Australia. On the other hand, Huang et al. [29] used long time series to estimate crop phenology. After analyzing different crops and VI in their work, the conclusion reported was that estimates made with NDVI showed the smallest differences concerning the control points. NDVI is an indicator of biomass greenness, is correlated with wheat biomass and grain yield [31], and is the most widely used and well-known VI [32]. Vallentin et al. [33] used a 13-year time series to study the relationship between some crop yields with different vegetative indices. The conclusion obtained from the analysis of several satellites was that higher-resolution satellites, such as Rapid Eye or Sentinel-2 performed better than lower resolution satellites. After studying different vegetative indices, they concluded that NDVI was one of the best for estimating wheat yield. However, deciding which VI is appropriate is not straightforward because several can be calculated by combining different spectral bands [34].

The main problem in using the NDVI is its tendency to saturate when the leaf area index (LAI) of the crop exceeds the value of 3 [35]. Theoretically, for more developed crops, which have accumulated high chlorophyll concentrations, the normalized red-edge difference (NDRE) is a better indicator of vegetation health/vigor than NDVI. The underlying reason is that red-edge light is more translucent to leaves than red light and is, therefore, less likely to be fully absorbed by a canopy. However, different studies have shown that the relationship between grain yield and NDVI is equal to or better than that shown with NDRE [21,36,37]. Another problem with NDVI is its sensitivity in the presence of bare soil. Therefore, other indices have been proposed to solve this problem, such as the soil-adjusted vegetation index (SAVI). However, to use this index, it is necessary to calculate the fraction of soil not covered by vegetation (L), which is not straightforward over large areas. Normally, the L value is replaced by the 0.5 value. The modified soil-adjusted vegetation index 2 (MSAVI-2) was developed to avoid the necessity of calculating the L value. However, recent studies [38,39] have shown that NDVI shows a better relationship with grain yield than MSAVI-2. Even knowing its limitations, the selection of NDVI is based on its high prevalence in the literature and its good relationship with wheat grain yield.

One of the challenges of modern agriculture is to adapt fertilization to the nutritional needs of crops to avoid nutrient losses and improve profitability. In Western Europe, a soil test to estimate available mineral N (ammonium plus nitrate) has been widely used to contribute to the determination of N to be applied. The optimum N rate is estimated by subtracting the available soil N from the N required by the crop [40].

Regarding the soil N availability, due to the spatial and temporal variability of Nmin, obtaining a representative sample of the plot is complicated, therefore, its application at the plot level is not considered very accurate [41]. Authors such as Ilseman et al. [42] did not find spatial dependence between Nmin values collected in the same plot. The Association of German Agricultural Research Institutes recommends taking 15 soil samples every 90 m$^2$ to capture Nmin variability, which supposes an unaffordable sampling intensity. In addition, the soil Nmin value can change rapidly depending on mineralization, leaching, crop extraction, and gaseous N losses. In the case of wheat, the soil Nmin should be determined before the end of winter, at the time the highest dose of fertilizer is applied, when the crop is active and growing rapidly. Therefore, soil sampling and analysis should

be made in a short period of time. The logistics required and the high cost of sampling and analyzing make this unfeasible, especially for a low value-added crop such as wheat. In addition, previous trials conducted in the study area showed that Nmin values at the beginning of winter were low [43], always being below 60 kg N ha$^{-1}$.

Given the difficulties involved in the soil Nmin sampling as well as the low N supply of the soil in the edaphoclimatic zone of the experiment, the main objective is to analyze whether the NDVI can be a good option to delineate SSMZ for fertilization since NDVI is a good way to predict the yield and subsequently, the N extraction by the plant. The next step will be to calculate an accurate fertilization rate for each SSMZ.

The procedure followed to achieve this goal is to divide the plot into NDVI homogeneous management zones and correlate them with zones based on a yield monitoring map. However, to be of practical application, the concordance between both site-specific management zones must be established before top dressing fertilization, which, in the study area, is applied at the beginning of the stem elongation phase. If NDVI's ability to delineate fertilization management zones is established, it could help farmers fertilize closer to crop needs.

For this purpose, 13 data sets (a total of 4517 yield measurements) of wheat collected in 2019 were analyzed. In total, 15 cloud-free Sentinel-2 images that cover wheat development were downloaded from which NDVI was calculated. Furthermore, geomorphological data, such as elevation, slope, and coefficient of variation (CV) of the slope were also included in the analysis. In the first step, the agreement between the classified yield maps (map formed by categorical variables: high and low production) and the classified NDVI (high and low NDVI) maps was compared. Finally, as some variables were not independent, a MANOVA analysis and the subsequent Descriptive Discriminant Analysis were performed to better analyze the data.

## 2. Materials and Methods

### 2.1. Study Area

This study was carried out with data from 13 wheat plots from two commercial farms located in the province of Araba/Álava (Figure 1a). All plots were sown at a seed rate of 230 kg ha$^{-1}$ between 19–25 November 2018 (Table 1). The same fertilization scheme was applied for all plots. The basal application of fertilizer was 53 kg N ha$^{-1}$, 36 kg P ha$^{-1}$ and 102 kg K ha$^{-1}$, and was applied on 30 December 2018. The second and third N top dressing fertilizer applications were made on 26 February 2019 and 25 March 2019. Calcium ammonium nitrate (ANC), which has a nitrogen concentration of 27%, was used for this purpose (Table 1). The ANC fertilizer dose applied was 220 and 210 kg ha$^{-1}$. The total N rate was 169 kg N ha$^{-1}$.

**Table 1.** Sowing and fertilization carried out on the study plots, specifying date, the product/wheat variety, and dose applied.

| Labor | Date | Variety/Product | Dose (kg ha$^{-1}$) |
|---|---|---|---|
| Sowing | 24 November 2018 | Filon | 230 |
| Fertilization | 30 December 2018 | Blending (13, 20, 30) | 410 |
| Fertilization | 26 February 2019 | ANC | 220 |
| Fertilization | 25 March 2019 | ANC | 210 |

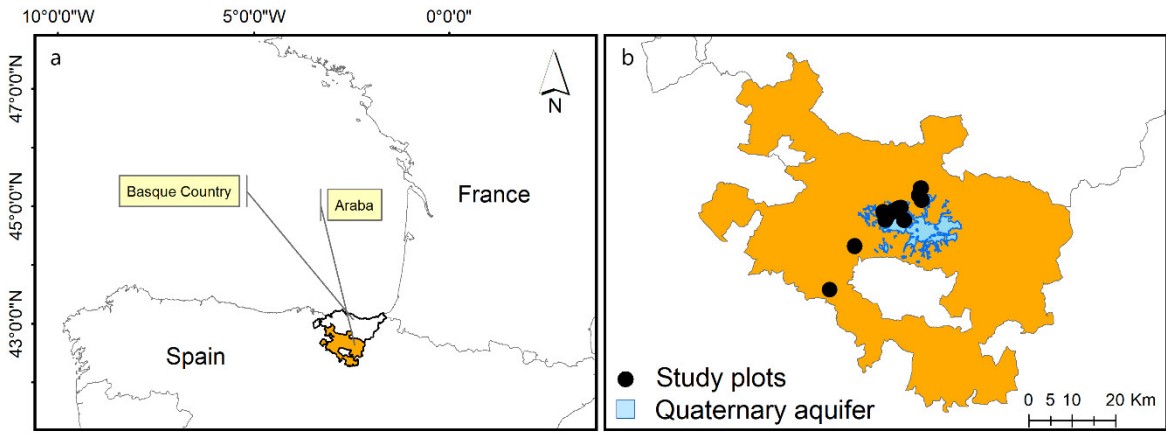

**Figure 1.** (**a**) Location of Araba/Álava province in Spain, (**b**) Geolocation of the 13 study plots in both regions, "Valles Occidentales" and "Llanada Alavesa".

The size of the plots range from 2.5 ha in Prado to 12.2 ha in Torres with an average plot size of 4.4 ha (Table 2). In total, 58.3 ha were analyzed. Table 2 shows some geomorphological properties of the plots, such as average elevation that ranges between 501 m of Prado to 554 m of Ollavarre, with the average elevation of all plots being 521 m. Moreover, the Prado plot is the flattest, with an average slope below 1%. However, Torres plot is the most sloping plot at 9%. Likewise, Torres plot is the most irregular, as it has a slope coefficient of variation (CV) of 6.8%, which is higher than the rest of the plots. On the other hand, the most regular plot is Foronda, with a CV for the slope of 0.52% (Table 2).

**Table 2.** Mean yield, area, number of yield sample points into each plot, and the corresponding soil type. Elevation, slope, and slope coefficient of variation have been obtained from Geoeuskadi (www.geo.euskadi.eus) spatial data repository (accessed on 9 of June 2022).

| Plot | Yield (t ha$^{-1}$) | Area (ha) | Nº. Sample Points | Soil Type | Elevation (m) | Slope (%) | CV (%) of Slope |
|------|------|------|------|------|------|------|------|
| Alto | 8.6 | 5.1 | 426 | Quaternary | 502 | 1.03 | 1.3 |
| Apelarri | 7.8 | 2.6 | 207 | Quaternary | 508 | 1.09 | 0.6 |
| Babea | 6.7 | 3.8 | 323 | Quaternary | 521 | 6.44 | 2.94 |
| Baratua | 5.6 | 2.7 | 217 | Quaternary | 511 | 1.24 | 0.87 |
| Foronda | 6.4 | 3.2 | 254 | Quaternary | 513 | 0.95 | 0.52 |
| Iruleku | 7.4 | 4.1 | 346 | Cretaceous | 534 | 2.45 | 2.39 |
| Kukura | 6.3 | 5.0 | 417 | Quaternary | 508 | 1.53 | 0.82 |
| Menor | 5.2 | 4.6 | 358 | Cretaceous | 538 | 4.12 | 1.43 |
| Ollavarre | 4.6 | 4.3 | 353 | Cretaceous | 554 | 6.12 | 2.67 |
| Otatza | 6.3 | 3.0 | 246 | Cretaceous | 541 | 4.46 | 1.66 |
| Parque | 4.7 | 5.2 | 246 | Cretaceous | 531 | 5.83 | 2.67 |
| Prado | 7.1 | 2.5 | 208 | Quaternary | 501 | 0.91 | 0.56 |
| Torres | 7.1 | 12.2 | 916 | Cretaceous | 511 | 9.53 | 6.83 |

Table 2 shows the main soil type where the study plots were established. To facilitate their interpretation, they will be referred to as Cretaceous and Quaternary soils. The soil formation in the area was strongly influenced by lithology, and in this case, corresponds to materials coming from the cretaceous and quaternary geological eras. In general, plots located over Cretaceous soils were steeper and more irregular, and the mean elevation was higher. Cretaceous soils were shallower (70 ± 20 cm), had higher $CaCO_3$ concentration (>50%), and their textures were very silty, with silt contents above 40%. The water retention capacity was lower (<100 mm) than in quaternary soils. In general, quaternary soils were characterized by being deeper (+120 cm), with a higher water retention capacity (165 ± 39 mm) and a lower (<25%) concentration of calcium carbonates ($CaCO_3$). The soil texture is loamier than in Cretaceous soils and have a higher stone content [44].

Twelve fields are located in the "Llanada Alavesa" region, near Vitoria, the capital of the province, while the other field is located in the west of the province, in the region known as "Valles Occidentales" (Figure 1b). The average elevation of study plots ranked between 502 and 554 m above sea level. Precipitation averages about 750 mm, with July and August being the driest months, both with a monthly rainfall of less than 50 mm (Figure 2). Summers are mild (20 °C) due to cold ocean currents, while winters are milder (6 °C) than in other climates of similar latitudes. According to Köppen [45], the climate of both regions is a "temperate oceanic" (Cfb). The average rainfall during the wheat-growing period was 474 mm, with January and May being the wettest months (Figure 2, blue line). Some of the study parcels are located over the Quaternary Aquifer of Vitoria (Figure 1b).

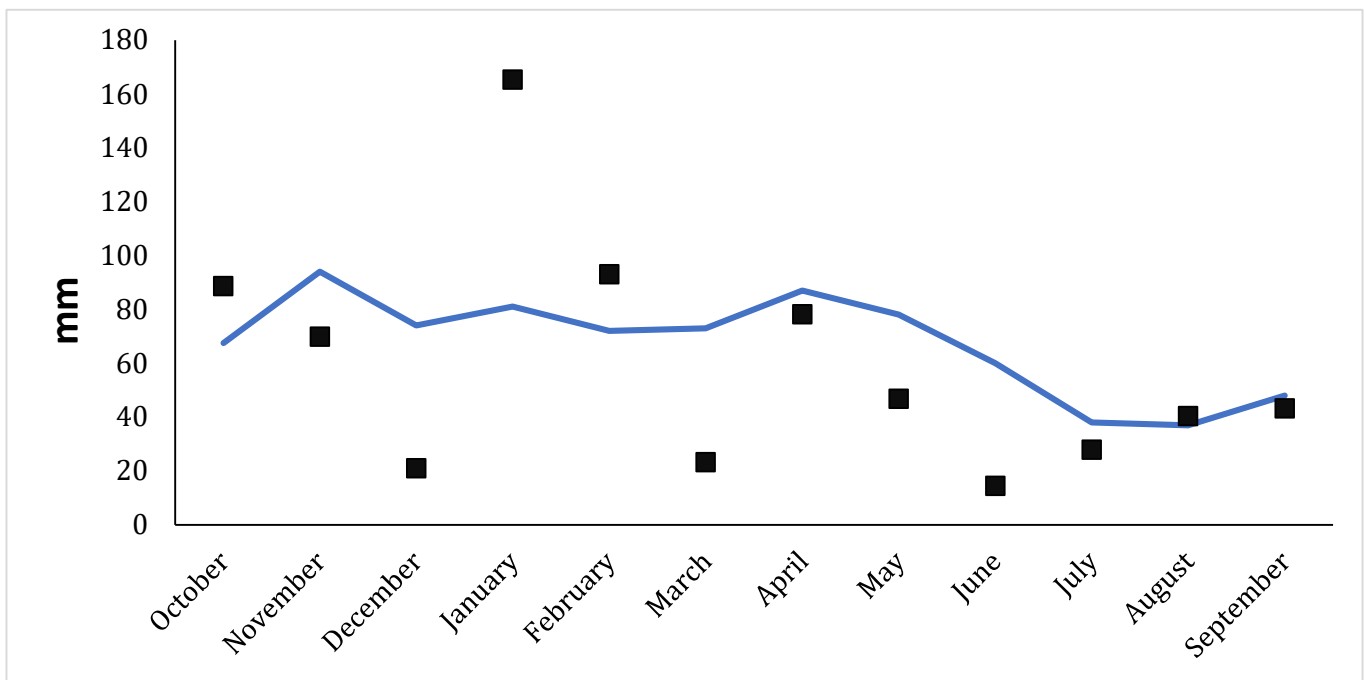

**Figure 2.** Vitoria-Gasteiz climatic station water precipitation regime. The blue line shows the average precipitation (mm) for the last 30 years (1989–2018). The black dots indicate 2018/2019 monthly precipitation (mm).

### 2.2. Yield Data

High-resolution yield data were obtained from a combination of a yield monitor with a GPS installed on a John Deere T560 harvester. The GPS receiver works with RX corrections, allowing it to work with a 15 cm precision. During the harvest period of 2019, between 23 July and 9 August, wheat yield data were acquired. Before it can be used, yield data need to be pre-processed to remove inaccurate grain yield measurements. The steps followed to pre-process the raw data are described below. In Figure 3, each yield (t ha$^{-1}$) measurement is represented through one point. The points that do not meet the requirements established in each pre-processing step are highlighted in red, to be subsequently removed. In the first step, measures with wrong latitude/longitude (e.g., outside the study plot) were eliminated (Figure 3(1)). In the following steps, some statistical procedures were then applied to remove values out of the criteria.

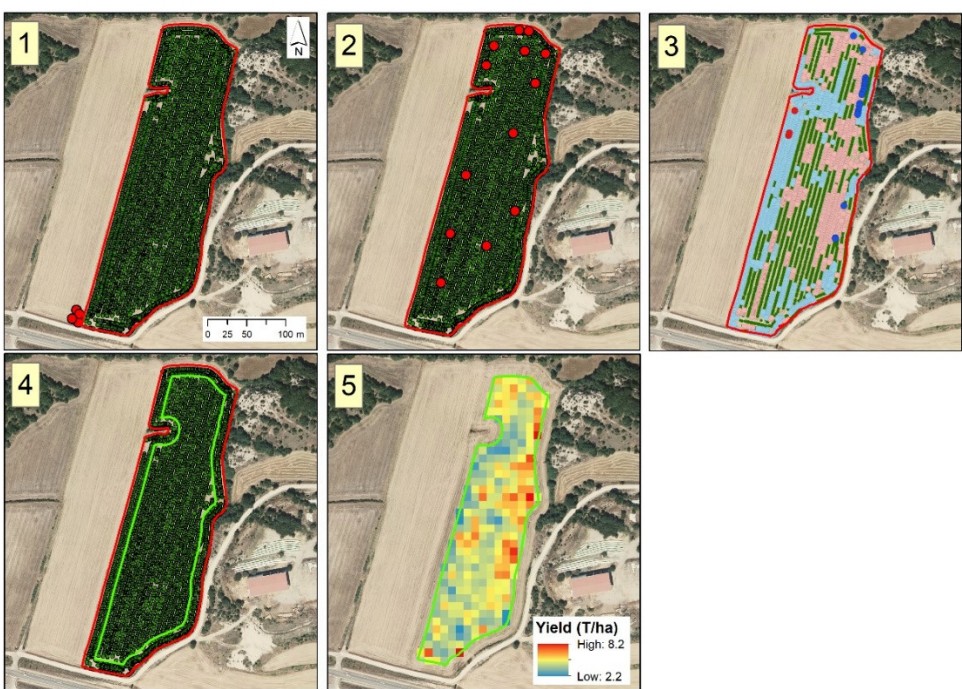

**Figure 3.** Preprocessing of yield monitor data followed in all plots. Example: Ollavarre plot: (**1**) remove points out of the plot, (**2**) remove points out of the stablished threshold, (**3**) remove outliers and Anselin-Moran local outliers (red/blue), (**4**) remove points out of the green buffer, (**5**) interpolate yield values.

The measurements eliminated were those with a moisture concentration below 8%, as well as values recorded with inadequate machine speed (i.e., when the distance between measurements was 0.8–1.6 m). In the next step (Figure 3(3)), outliers were removed. For this purpose, the methodology described by Taylor et al. [46] was applied. Additionally, to identify and eliminate spatial outliers, the Local Moran 1 [47] test was performed. This test identifies measurements of high values surrounded by low values and vice versa. Figure 3(4) shows the limit of the safety buffer set on each plot (represented with a green line). To ensure that all Sentinel-2 pixels were completely within the study plot, all pixels between the green line and the plot boundary were removed. This procedure ensures that the possible distortion caused by the edge effect was minimized. In the final step (Figure 3(5)), each semivariogram was adjusted to the corresponding plot. Then, the Ordinary Kriging method was used to interpolate the yield points into a continuous yield map. New yield maps were resampled to 10 × 10 m resolution and adjusted to align with Sentinel-2 pixels. Finally, to extract the information (NDVI and grain yield values), sample points were generated in the center of each interpolated raster cell, producing a sample dataset containing 4517 values. Table 2 shows the number of Sentinel-2 pixels corresponding to each study plot. With 916 pixels, Torres is the plot with more pixels, while Apelarri has the lowest number of pixels at 207. Thus, the average number of pixels of the plots was 347. Table 2 shows the mean yield values (t ha$^{-1}$) and the number of sample points after finishing the pre-process.

*2.3. Sentinel-2 Vegetation Index Data and Growth Stages*

The Sentinel-2 mission is composed of twin satellites launched by the European Space Agency (ESA) in June 2015 and March 2017. They provide thirteen-band multispectral images with a revisit interval of 5 days. The satellite data were freely downloaded from the "Copernicus Open Access Hub" platform (https://scihub.copernicus.eu/) (accessed on 9 June 2022) as an L2A product, so they have been atmospherically corrected [48]. As the 30 TWN mosaics covered all the study plots, the whole wheat development

cycle was covered by downloading 15 cloud-free images (Table 3). The selected bands were NIR (B8: 842 nm, bandwidth: 106 nm) and red (B4: 665 nm, bandwidth: 31 nm), both having 10 × 10 m spatial resolution. The NDVI was calculated using the following equation:

$$\text{NDVI} = \frac{\text{NIR} - \text{RED}}{\text{RED} + \text{NIR}} \tag{1}$$

**Table 3.** Sentinel-2 T30TWN tiles used in this work and the captured dates.

| Data | Sentinel-2 Tile |
|---|---|
| 4 February | S2B_MSIL2A_20190204T110439_N0211_R094_T30TWN |
| 8 February | S2A_MSIL2A_20190208T110221_N0211_R094_T30TWN |
| 13 February | S2A_MSIL2A_20190213T110149_N0211_R094_T30TWN |
| 19 February | S2A_MSIL2A_20190218T110111_N0211_R094_T30TWN |
| 23 February | S2B_MSIL2A_20190223T110039_N0211_R094_T30TWN |
| 5 March | S2A_MSIL2A_20190228T110001_N0211_R094_T30TWN |
| 15 March | S2B_MSIL2A_20190315T105819_N0211_R094_T30TWN |
| 20 March | S2A_MSIL2A_20190320T105741_N0211_R094_T30TWN |
| 30 March | S2A_MSIL2A_20190330T105631_N0211_R094_T30TWN |
| 29 April | S2A_MSIL2A_20190429T105621_N0211_R094_T30TWN |
| 14 May | S2B_MSIL2A_20190514T105629_N0212_R094_T30TWN |
| 3 June | S2B_MSIL2A_20190603T105629_N0212_R094_T30TWN |
| 8 June | S2A_MSIL2A_20190608T105621_N0212_R094_T30TWN |
| 18 June | S2A_MSIL2A_20190618T105621_N0212_R094_T30TWN |
| 28 June | S2A_MSIL2A_20190628T105621_N0212_R094_T30TWN |
| 18 July | S2A_MSIL2A_20190718T105621_N0213_R094_T30TWN |

Four key growth stages (Zadoks scale [49]) were identified to analyze the evolution and facilitate the understanding of the paper: tillering (GS 20), stem elongation (GS 30), heading (GS 61), and maturation (GS 87). The onset of the four key growth stages was determined using STICS (Simulateur mulTIdisciplinaire pour les Cultures Standard) software [50], as Zadoks stages could not be measured in the plots. STICS uses the number of hours above 0 to estimate the passage from one to the other [50].

*2.4. Geomorphological Variables: Elevation, Soil Type, and Orthophoto*

The Digital Elevation Model (DEM), derived from the LiDAR flight of 2016, was used to calculate geomorphological variables (Table 2). This information can be found in Geoeuskadi [51], Basque Country's official repository of spatial information. The main soil type was determined using a Basque Country lithological map (1:25,000).

Finally, the 2009 orthophoto from the Geoeuskadi repository was also used. With a pixel resolution of 25 × 25 cm, the orthophoto shows the plots with bare soil.

*2.5. Topographic Wetness Index (TWI)*

The topographic wetness index (TWI) is a popular and widely used index to infer information about the spatial distribution of moisture conditions in an area [52]. The local morphological analysis allows for identifying zones with a high capacity to accumulate water [53].

The calculation of TWI is usually based on a gridded DEM (Equation (2)):

$$\text{TWI} = \ln(a/\tan\beta) \tag{2}$$

where "a" is the upslope contributing area per unit contour length (or Specific Catchment Area, SCA) and $\tan\beta$ is the local slope gradient for estimating a hydraulic gradient. The DEM used to calculate the TWI has a pixel resolution of 5 × 5 m.

*2.6. Data Analysis*

2.6.1. ISODATA

Plots were analyzed individually, so values for the high and low zones may vary between plots. Considering the size of the plot and the fact that the objective of these zones is to create site-specific management zones (SSMZ) to adjust the fertilization, it is not possible to divide them into more than two zones. Once the different homogeneous zones are created, the classified NDVI and grain yield maps can be compared between them. The procedure for defining the SSMZ is based on an iterative algorithm that starts by arbitrarily assigning a mean to each class. Pixels are then reallocated to each group based on the minimum Euclidean distance between each pixel value and the mean value of each group. Each class is established when the maximum number of iterations is reached, or when the number of pixels changing from one class to another does not exceed a preset threshold [54].

2.6.2. Kappa Index (KI)

The Kappa Index (KI) is a statistic used to measure inter-rater reliability for categorical variables [55]. The degree of similarity between maps is quantified using KI, as it is considered a more reliable measure of agreement than simple percent agreement, since KI takes into account the likelihood of agreement occurring by chance [55]. A minimum concordance of 0.40, (moderate agreement [56]) between yield maps and classified NDVI images was stablished to use the NDVI as a tool to delineate the fertilization SSMZ. A negative Kappa Index value represented worse agreement than expected [57]. Low negative values (0 to −0.10) may generally be interpreted as "no agreement". The more negative the index, the lower the degree of agreement between the maps [58].

2.6.3. MANOVA Test

Multivariate analysis of variance (MANOVA) was used to detect whether plot morphological properties or differences in development influence the agreement between NDVI and yield. Performing multiple ANOVAs for different NDVI dates can lead to misleading and inconsistent results, as the time series data are not independent. Although the MANOVA method is slightly more complex than ANOVA, it has given good results when used to define management areas [59].

In addition, the canonical correlation indicates the proportion of the variation in the model that is explained by the grouping variable [60], which would be NDVI of different data or plot morphological properties. Once the MANOVA shows significant differences, the capacity of each variable to differentiate the groups can be analyzed. Thomas, in [61], proposed descriptive discriminant analysis (DDA) as a "post hoc" analysis for this purpose. DDA was previously used in precision agriculture with successful results [62]. Briefly, the standardized discriminant function coefficient (SDFC) and structure coefficients (SC) were used to interpret the DDA. The SDFCs measure the contribution of each variable to the discriminant function, while the SCs measure the correlation between the discriminant function and the variables. Finally, by multiplying the SDFC and the SC, the parallel relationship coefficient (parallel RDC) can be obtained, which allows the evaluation of the contribution of each variable to differentiate between the two groups [61]. In this study, the two groups would be plots with agreement and plots without agreement between grain yield and NDVI.

In summary, the MANOVA method and subsequent DDA were used to understand which properties were most decisive in discriminating between groups.

An overview of different steps of the data acquisition procedure and analysis is given in Figure 4.

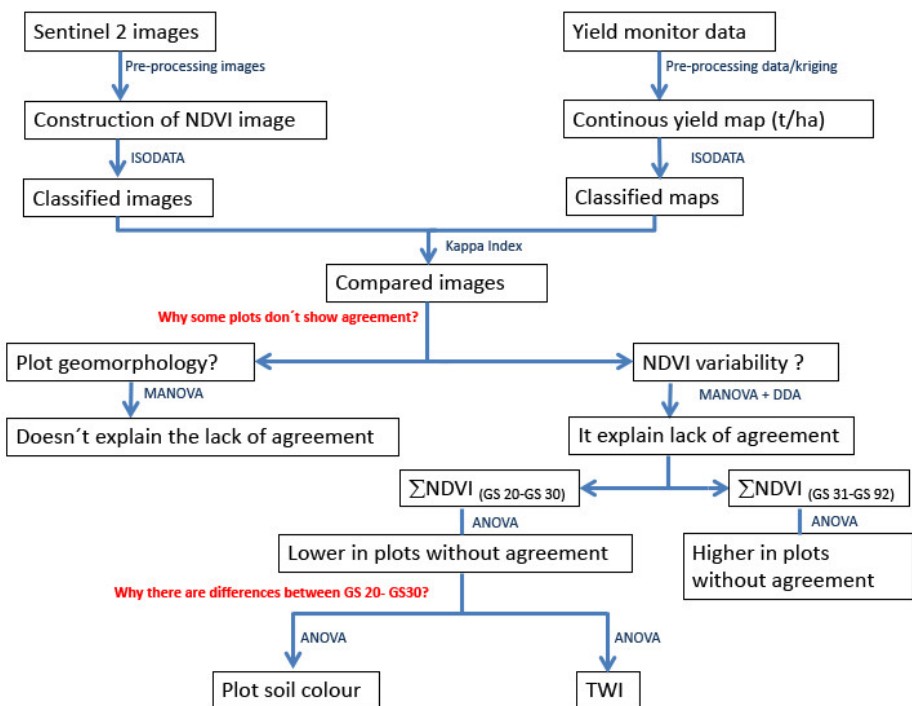

**Figure 4.** Flowchart of the steps carried out to analyze the data of the study. Out of the boxes, in blue, statistical procedures used are specified. In red are the two critical questions that need to be answered.

## 3. Results and Discussion

### 3.1. NDVI Evolution

Figure 5 shows the evolution of NDVI in the 13 plots. The NDVI index was selected as a tool to delimit early SSMZ because it has interesting properties related to grain yield. NDVI is an indicator of the combined effects of chlorophyll concentration, canopy leaf area, and yield. In addition to being easy to measure, it does not involve sample destruction. Plots started being monitored once the crop covered most of the soil, since bare soil can alter NDVI values [63]. In all of them, the maximum value was reached during the stem elongation phase, on 28 April, just after the second fertilizer application (Figure 5). After 18 June, the end of the heading phase, NDVI values dropped rapidly, indicating the onset of senescence [64]. On 13 February, a measurement error could have occurred, as the NDVI value dropped instead of increasing, and no apparent reason for this behavior has been found. Therefore, data from this date were not used in the subsequent analyses. Although the NDVI value on 23 February was lower than on 19 February, this measurement was not considered an error because irradiance and temperature were high on the previous days. As a consequence, the crop had a strong growth, producing a demand for N, stressing the plant, and this was reflected in crop vigor. The farmer made the first fertilizer application on 26 February (Table 1), and from this date, the NDVI value increased until reaching a maximum on 28 April. As the NDVI values corresponding to 18 July reflect that the wheat had already dried (Figure 5), these values were also excluded from the subsequent analysis.

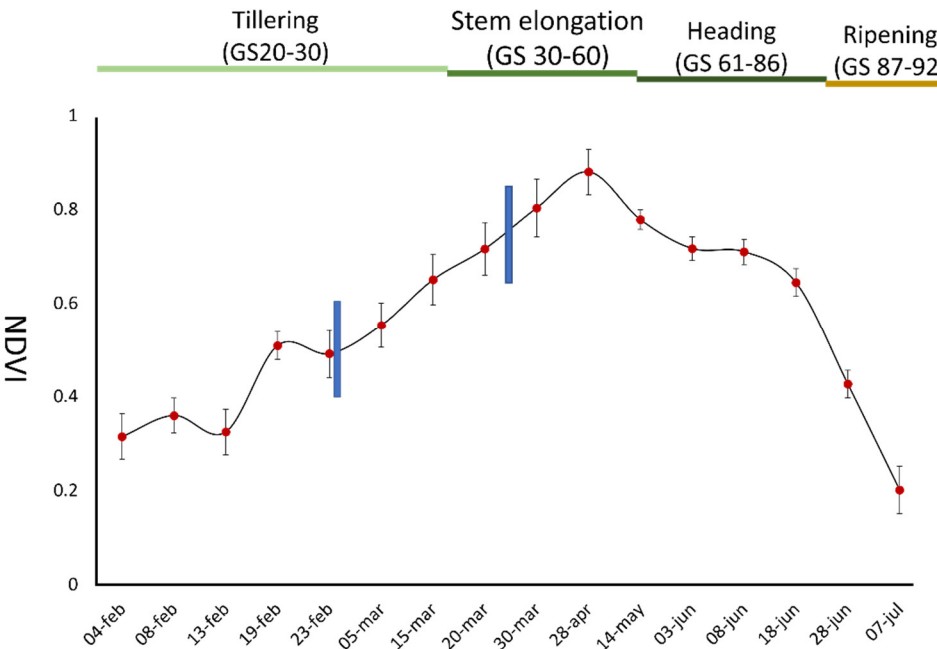

**Figure 5.** Seasonal evolution of NDVI across the 13 studied plots in year 2019. Red dots are the mean value of NDVI, calculated using the 13 mean plot NDVI values, and error bars represent the standard deviation. In the figure, the onset and duration of the four key phenological stages are represented. The vertical blue lines represent the two dates when ANC fertilizer was applied, 26 February and 25 March.

### 3.2. Comparison of Yield Maps and Temporal NDVI Images

The average yield ranged from 4.65 t ha⁻¹ in Ollavarre to 8.66 t ha⁻¹ in Alto, with an average production of 6.50 t ha⁻¹ (Table 2). Considering that the plots analyzed are conventionally managed and cover different geomorphological zones, the dataset can be considered representative of the "Llanada-Alavesa" (Figure 1).

In areas where NDVI is higher, higher wheat yields are expected [65]. Therefore, in areas with a high NDVI, the crop demands more N. In the study area, blanket fertilizer is applied during the early stem elongation phase (GS 30) [66]. Generally, wheat reaches this growth phase in late March or early April. Therefore, for NDVI to be an effective tool for delineating fertilizer management zones, it is necessary to confirm that the agreement between NDVI and grain yield is moderate (KI > 0.40) at this growth stage.

Table 4 shows that 9 of the 13 plots achieve at least moderate agreement (KI > 0.40) between NDVI and yield. Four plots (Babea, Baratua, Iruleko, and Kukura) reached a moderate agreement during February or March (tillering, GS 20), while Menor and Otatza plots exceeded the threshold during April (stem elongation, GS 30). Finally, Alto, Foronda, and Torres parcels reached this threshold during May (flowering initiation, GS 61). However, Apelarri, Ollavarre, Parque, and Prado did not achieve the required agreement during any growth phase. Figure 6 shows the differences between a plot (Babea) with high KI (KI = 0.7) and one with low KI (KI = 0.34). To facilitate the comprehension, the group formed by Babea and eight other plots that show an agreement will be referred to as "related" while the rest will be referred to as "unrelated".

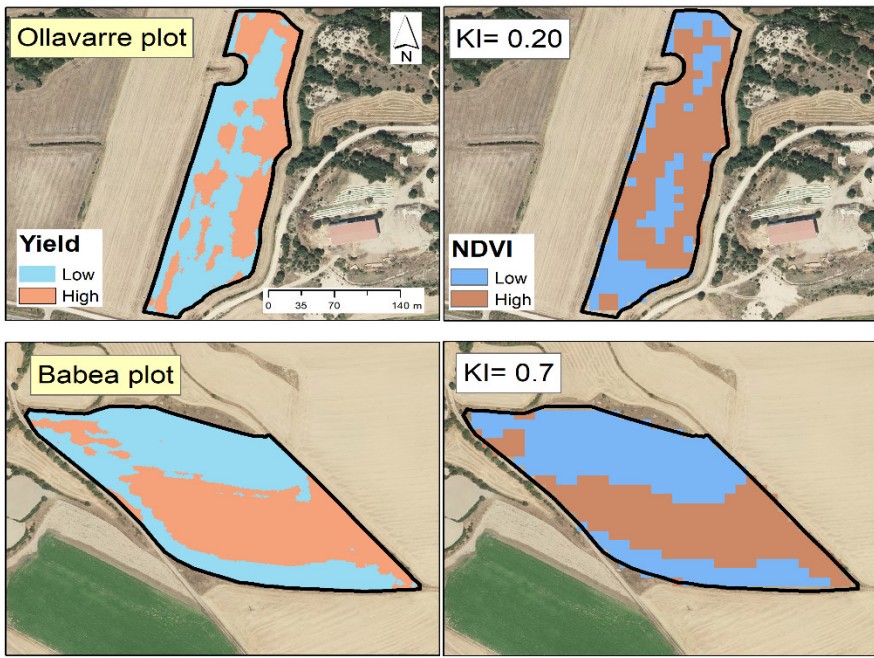

**Figure 6.** NDVI and yield classified maps, and agreement between them according to Kappa Index (KI). Plot without concordance (**top**) and plot with concordance (**bottom**).

**Table 4.** Values of Kappa Index between yield map and NDVI images of different dates in 2019 for 13 wheat plots. In bold is the first date when a moderate agreement is achieved.

| Data | Alto | Apelarri | Babea | Baratua | Iruleku | Kukura | Menor | Ollavarre | Otatza | Parque | Prado | Torres | Foronda |
|---|---|---|---|---|---|---|---|---|---|---|---|---|---|
| **4 February** | - | - | - | 0.26 | 0.32 | - | −0.1 | 0 | 0 | 0 | −0.16 | −0.04 | −0.26 |
| **8 February** | −0.24 | - | 0.2 | 0.17 | - | 0.32 | - | −0.27 | 0.16 | 0.18 | −0.13 | −0.09 | 0.19 |
| **19 February** | −0.31 | −0.23 | 0.39 | 0.29 | - | - | - | −0.27 | 0 | 0.19 | −0.17 | −0.01 | - |
| **23 February** | −0.26 | - | 0.3 | 0.29 | 0.24 | 0.27 | - | −0.29 | 0.14 | - | −0.26 | −0.06 | - |
| **5 March** | −0.19 | −0.12 | 0.55 | **0.4** | **0.62** | **0.46** | - | −0.25 | - | - | −0.16 | 0.07 | - |
| **15 March** | - | - | 0.55 | 0.44 | - | 0.52 | - | −0.22 | - | - | −0.12 | 0.07 | 0.28 |
| **20 March** | - | −0.12 | - | - | - | - | - | - | - | - | - | - | - |
| **30 March** | 0.15 | - | 0.41 | 0.56 | - | 0.45 | - | - | 0.35 | 0.3 | −0.11 | 0.15 | - |
| **28 April** | 0.21 | - | 0.66 | 0.52 | 0.17 | 0.44 | **0.6** | 0.17 | **0.42** | 0.21 | 0.08 | 0.12 | - |
| **14 May** | 0.27 | - | 0.55 | 0.51 | - | 0.44 | 0.55 | 0.2 | - | 0.13 | 0.17 | 0.16 | 0.35 |
| **3 June** | 0.35 | 0.34 | 0.54 | 0.48 | 0.24 | 0.55 | 0.55 | 0.16 | 0.52 | 0.21 | 015 | 0.32 | **0.55** |
| **8 June** | 0.33 | 0.32 | 0.7 | 0.45 | 0.25 | 0.5 | 0.63 | 0.16 | 0.5 | 0.31 | 0.10 | **0.39** | 0.61 |
| **18 June** | **0.39** | 0.31 | 0.7 | 0.5 | 0.25 | 0.58 | 0.7 | 0.19 | 0.49 | 0.32 | 0.11 | 0.56 | 0.62 |
| **28 June** | 0.33 | 0.32 | 0.65 | 0.43 | - | 0.63 | - | 0.15 | 0.45 | 0.26 | 0.05 | 0.34 | 0.62 |

In almost all plots, the agreement between NDVI and grain yield increased until the end of the heading phase (GS 86) (Table 4). The maximum values of the agreement were reached during the third week of June, which shows that the results are similar to those published by Hunt et al. [66] in the United Kingdom and Martí et al. [67] in Spain. Similar results were obtained in the study by Aranguren et al. [21], where the relationship between wheat yield fertilized with different doses and NDVI was analyzed. Finally, Royo et al. [68] and Babar et al. [69] observed that the highest relationship between NDVI and grain yield was established during anthesis or the milky grain stage, and reported that it was maintained until the ripening stage (GS 80).

However, the Iruleko plot was the exception (Table 4) since it reached the maximum KI (0.62) on 5 March, during the tillering phase (GS 30). Other authors [31], who worked with two plots and multiple fertilizer doses, reported a small peak correlation between NDVI and grain yield during the tillering stage (GS 30) ($R^2 = 0.25$). After that, the

correlation decreases until the end of the flag leaf stage (GS 39) ($R^2 = 0.03$). Afterwards, the correlation increases until ripening (GS 80) ($R^2 = 0.55$). Naser et al. [70], in their 2020 publication, supported this information as they reported a similar pattern between grain yield and NDVI.

### 3.3. *Analysing the Lack of Agreement between Yield and NDVI in Some Plots*

#### 3.3.1. Multivariate Analysis of Plot Geomorphology

Part of the variability in grain yield could be explained by the geomorphological variability of the plots, as these properties can influence plant development and, ultimately, grain yield [71,72]. Climate influence has not been considered in this study because all plots are near each other.

A MANOVA analysis was performed to confirm whether the source of the lack of agreement could be soil geomorphology. This analysis was chosen as some of the variables are correlated (Slope and Slope CV). Plots that show agreement between grain yield and NDVI were compared with plots that do not show agreement (Table 5). The MANOVA did not show significant differences ($p > 0.05$). Therefore, it was confirmed that plot geomorphology was not the source of the lack of agreement.

**Table 5.** Mean values and standard deviation of the geomorphological properties of the 13 plots. The values have been grouped considering the agreement with the NDVI.

| Properties | Unrelated Group | Related Group |
|---|---|---|
| Plot number (n) | 4 | 9 |
| Elevation (m) | 519.88 ± 14.3 | 523.50 ± 24.3 |
| Slope (%) | 3.55 ± 3.0 | 3.48 ± 2.8 |
| Slope CV (%) | 2.08 ± 1.9 | 1.62 ± 1.2 |

#### 3.3.2. Analysis of NDVI Value using MANOVA and DDA

A new MANOVA was used to analyze if there were differences in NDVI between related and unrelated plots. For this purpose, NDVI values between 4 February (GS 21) and 28 June (GS 87) were used (Table 6). The MANOVA showed significant differences ($p < 0.001$) between the two groups. Therefore, to determine which NDVI dates contributed most to differentiate both groups, a DDA was performed (Table 6). The canonical correlation derived from the DDA explains 84% of the variability. Therefore, the model resulting from grouping variables is a good representation of reality.

**Table 6.** Results of DDA after MANOVA being significative, showing contribution of NDVI values from different data to classification of the plots on the group with or without agreement among NDVI and yield.

| Phenological Stages | Vegetation Index | Date | Discriminant Analysis (DDA) | | |
|---|---|---|---|---|---|
| | | | SDFC | SC | Parallel DRC |
| Tillering (GS 20–G S30) | NDVI | 4 February | 0.32 | 0.16 | 0.05 |
| | NDVI | 8 February | −1.61 | 0.08 | −0.13 * |
| | NDVI | 19 February | 2.92 | 0.34 | 1.01 * |
| | NDVI | 23 February | −0.60 | 0.23 | −0.14 * |
| | NDVI | 5 March | −0.65 | 0.24 | −0.15 * |
| | NDVI | 15 March | 1.58 | 0.20 | 0.32 * |
| | NDVI | 20 March | 0.09 | 0.17 | 0.02 |
| Stem Elongation (GS 31–GS 60) | NDVI | 30 March | −0.68 | −0.07 | 0.05 |
| | NDVI | 28 April | 0.05 | 0.10 | 0.01 |
| | NDVI | 14 May | −1.81 | −0.01 | 0.01 |
| Heading (GS 61–GS 86) | NDVI | 3 June | 1.06 | −0.02 | −0.02 |
| | NDVI | 8 June | −1.11 | −0.03 | 0.04 |
| | NDVI | 18 June | 0.71 | −0.03 | −0.02 |
| Ripening | NDVI | 28 June | −0.15 | −0.11 | 0.02 |

| (GS 87–GS 92) |

\* Indicates the variables that contribute most to differentiate between the two groups, standardized discriminant function coefficient (SDFC), and structure coefficients (SC).

Table 6 shows that the parallel DRC values between 8 February and 15 March are the closest to 1 or −1. Therefore, the NDVI values of these dates contributed the most to differentiate between the two groups. These dates encompassed the tillering growth phase. So, what happens during tillering to cause the dissociation between grain yield and NDVI?

### 3.3.3. Tillering, Dissociation among Yield Map and NDVI Images

Wheat grain yield is affected by nutrient uptake, metabolism, photosynthesis, respiration, carbon distribution, leaf senescence, and plant water conditions [73]. Usually, plants with larger vegetative development (until GS 60) have a higher yield [74] as they ensure a greater supply of carbohydrates for grain filling. Along with kernel number and weight, the number of spikes determined wheat grain yield [75]. The duration and intensity, as well as the stage of development at which the stress is applied, determine the extent of grain yield reduction [75]. At the beginning of the vegetative phase (GS 13–GS 30), the environment is the most influential factor for stem development. These early stems will later produce spikes with the highest likelihood of viability [76]. Usually, only tillers that grow before the wheat develops 4–6 leaves (GS 32) on the main stem develop fruitful spikes [77,78] since the rest are aborted before heading [79]. Therefore, tiller survival probability is affected by the sowing date [80]. Considering the previous information, it can be assumed that small changes in the tillering phase length can affect grain yield.

Figure 7 shows the different evolution of NDVI of the related plots compared to the unrelated. Generally, on unrelated plots, values of NDVI were lower until the start of stem elongation (GS 31). The Apelarri plot was the exception; although, the NDVI on February 4 was one of the lowest, it increased immediately and remained at the same values as the related parcels. The steeper slope of the NDVI in the unrelated plots during the tillering phase (GS 20–30) indicates a rapid increase of NDVI, which represents that the wheat in these plots has been less time to develop in this phase. After stem elongation (GS 31), NDVI values reached by Apelarri, Ollavarre, and Prado were among the highest. The differences in NDVI observed in Figure 7 between the related and unrelated plots were measured by means of an ANOVA. For this purpose, the more representative NDVI values of the DDA (highlighted in Table 6 with \*) were summed. These dates coincide with the dates when the NDVI of the unrelated plots are lower (Figure 7). The ANOVA showed that the sum of the average NDVI values during the tillering phase of unrelated plots was significatively lower (2.69) than the sum of the average of NDVI of the related plots (3.05). During the tillering, a lower NDVI value was indicative of a lower vegetative growth since the presence of bare soil decreases the NDVI value [81]. When the photoperiod increases, wheat progresses from tillering (GS 20) to stem elongation (GS 30). Therefore, a delay in the onset of tillering shortens the duration of this stage [81]. Then, another ANOVA was conducted by summing the NDVI values from 20 March (GS 31) to 28 June (GS 87). The sum of NDVI values of unrelated plots was slightly higher (7.63) than the related plots (7.59). Therefore, although there were initial differences in NDVI values between the two groups, crop vigor (NDVI) was equalized starting from the stem elongation growth stage (GS 31).

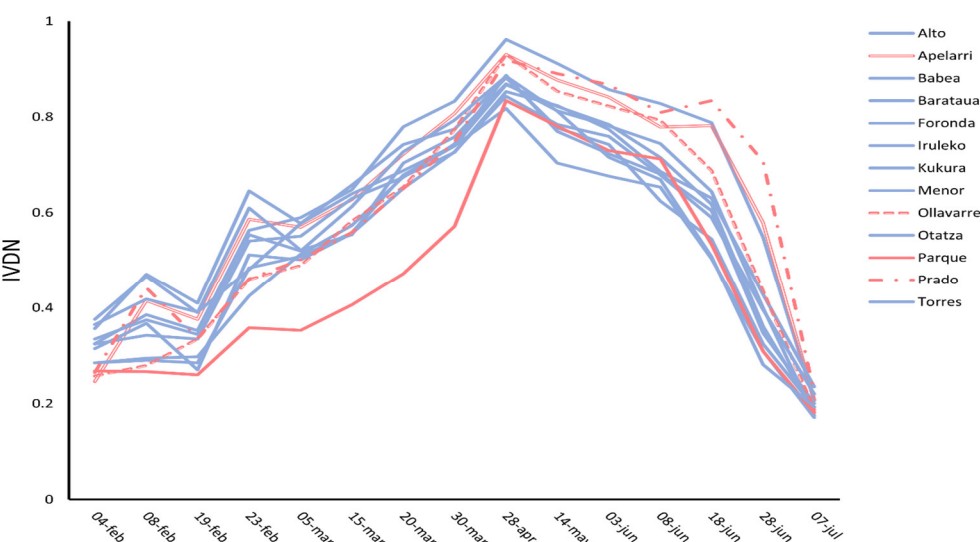

**Figure 7.** NDVI temporal evolution of the 13 plots. Blue color shows the plots that have reached a moderate level of agreement (related plots) between NDVI and yield. Pink color shows the group of plots that did not reach a moderate agreement (unrelated plots). The plots belonging to this group have been individually identified with different types of lines.

### 3.3.4. Why Some Plots Have a Lower NDVI during the Tillering Phase

A lower NDVI value during tillering is indicative of slower crop development. The factors that most influence the onset of tillering are the anaerobic state of the soil and temperature [82], both directly related to the rainfall regime. It is estimated that the tillering phase started at the end of January. January was unusually rainy, raining 165 mm, more than double the 30-year average of 82 mm (Figure 2). Although some soil moisture is necessary, excessive soil water accumulation can delay tillering initiation by stressing the crop. Thus, a saturation of soil air pores causes hypoxia or anoxia due to a total or very low absence of plant-available oxygen [83]. Low oxygen availability disturbs plant physiology and metabolism, resulting in reduced growth, retarded development and, finally, reduced grain yield [84].

The study plots are situated in a flat lowland area with low hydraulic gradients and a high groundwater table [85]. In addition, the phreatic level is close to the surface (0–1.5 m), so it is common for some areas of the plots to periodically become waterlogged [86]. Thus, plot topography influences the ability to drain and hold water. Therefore, due to the high rainfall produced in early 2019, zones with better drainage may have encouraged the onset of tillering. The topographic wetness index (TWI) considers the topography of an area to determine the capacity to retain or drain water. The ANOVA results show that the TWI values of the unrelated plots (TWI = 4.70) were significantly higher ($p < 0.01$) than related plots (4.42), which indicates that their water holding capacity is higher. Therefore, after a rainy January, water may saturate the soil pores.

The color of the soil affects its temperature directly since dark-colored soils absorb more radiant heat than light-colored ones [87]. Therefore, darker soils warm up more rapidly than pale soils [88]. The orthophoto of 2009 provides the opportunity to analyze the soil color of the study plots, as the soil was bare of vegetation when it was taken. A new variable was obtained by summing the values of the three channels (red, green, and blue) of the orthophoto. The values of the new variable vary between 0 (black) and 765 (white). Another ANOVA was performed to compare the soil color of related and unrelated plots. The analysis showed that unrelated plots are significantly ($p < 0.001$) lighter (mean $\sum$RGB 375.6) than related plots (mean $\sum$RGB 341.9).

In summary, soils of the unrelated plots are lighter and have a higher water holding capacity. The extraordinary rainfall period and the physical characteristics of unrelated

parcels (soil color and topography) delayed the onset of tillering, which was reflected in the lower NDVI values during the tillering phase. The work of Arguello et al. [89] discussed how waterlogging changes the importance of yield components of wheat. In the control treatment, the number of kernels spikes$^{-1}$, the number of spikes in each m$^{-1}$, and 1000 kernel weight were the components that most influenced the yield. Nonetheless, in wheat that had undergone waterlogging, the number of spikes m$^{-1}$ and kernel weight spike$^{-1}$ were the two most relevant components. In this study, the onset of tillering was delayed due to the waterlogging of some plots due to previous heavy rainfall.

Wheat suffered water stress from May onwards (GS 60) as it received less water. Therefore, it is understandable that the NDVI of the plots with higher water holding capacity matched with the related plots. This higher NDVI at the waterlogged plots from GS 30 on was already reported by [89] in a previous article. This suggests that, despite the lower number of tillers, the ears developed from these have developed adequately, and as a consequence of being less stressed, the other two yield components (kernel number spikes$^{-1}$ and 1000 kernel weight), which are more difficult to detect by remote sensing, have compensated for the lower number of spikes/m$^2$.

In the early stages of development, the number of tillers is well correlated with biomass, but this relationship decreases during crop development [90]. Considering that NDVI reflects changes in biomass better than changes in grain yield [31], it can be assumed that the greater the influence of the number of tillers per m$^2$ on yield, the better the relationship with NDVI [89].

## 4. Conclusions

In one-third of the plots analyzed, the NDVI showed moderate agreement (KI > 0.4) with the classified yield map before the time for the last N fertilizer application. Therefore, for these cases, the use of NDVI would be a useful tool to delineate yield site-specific management zones within the same season. However, for another third of the plots, a moderate agreement (KI > 0.40) was reached between NDVI and grain yield maps only after the last N fertilization. For the rest of the plots, a moderate concordance was not reached at any time of crop development. Therefore, for two-thirds of the plots, the NDVI would be useful to delimit yield-specific homogeneous management zones in the following seasons regarding N fertilization. This is feasible, provided that the agreement between NDVI and yield does not depend on the growing season.

An attempt has been made to provide an answer for plots that do not show agreement between NDVI classified images and grain yield classified maps. The mismatch may be the result of a delay in the onset of tillering, which causes a shortening of the tillering phase. As a result, the plant does not develop enough tillers, so it tries to compensate for this through other yield components (number of grains per tillers and grain weight) that are not well estimated by the NDVI index.

In summary, NDVI is a promising tool for establishing site-specific yield management zones. However, a better understanding of the underlying mechanisms is needed to develop a methodology applicable to farmers. Therefore, further studies are needed to increase knowledge on this issue.

**Author Contributions:** Conceptualization, methodology, software, data processing, formal analysis original draft preparation, visualization, investigation, interpretation, A.U.; methodology, resources, A.C.; methodology, writing—reviewing and editing, supervision of parameter computing, funding acquisition, project administration, A.A. All authors have read and agreed to the published version of the manuscript.

**Funding:** This work was funded by BIKAINTEK, grant of the Basque Government, Department of Economic Development, Sustainability, and Environment through the NITRALDA Project.

**Data Availability Statement:** Data are available in a publicly accessible repository that does not issue DOIs. The topographic data can be found in

https://www.geo.euskadi.eus/cartografia/DatosDescarga/, accessed on 8 June 2021. The satellite information data can be found in https://scihub.copernicus.eu/dhus/#/home, accessed on 9 June 2021

**Acknowledgments:** We are grateful to Javier Alava Hermanos Torre farmers for allowing us to conduct the research in their plots.

**Conflicts of Interest:** The authors declare no conflict of interest.

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
