# Peer review of "A First Approach to Determine If It Is Possible to Delineate In-Season N Fertilization Maps for Wheat Using NDVI Derived from Sentinel-2"

_remotesensing, doi:10.3390/rs14122872_

Round 1
Reviewer 1 Report
Dear authors
Thank you for your manuscript submission and your efforts to review and update it. This new version is significantly improved. Nothing more to comment on or say.
Reviewer 2 Report
Dear authors
I have both general and more specific comments to your manuscript.
general comments:
When I read your manuscript I cant find a good answer to your title. is it possible to delineate N fertilization using NDVI? I believe based on your results that the answer is no, do you agree?
It is not enough to be able to devide the fields into management zones, you also have to define how much N should be applied to each zone before it is useful for the farmer.
do a search and replace on upper cases like ha-1 which should be ha-1
x is replaced by -
More specific comments:
line 20 - define DDA
line 36 and 37 - is this the reason for increased use of N? I am not sure, I believe it is due to just want to have a higher yield, I would like you to comment this.
line 53 - PA doesnt necessarily increase yield and sustainability, you should be more critical to literature
lin 58 - avoid using etc.
line 83 to 85 - be critical to theses suggestions, is it always true?
line 101 and 102 - you need a reference to this
line 125 - define what you mean by 'classified'
line 131 - 11/24/2018 is not a week
line 131 - seed density, do you mean seed rate
line 133 - replace made with applied
line 174 - 6Co is the correct way to write it
figure 2 - title on y axis is wrong
line 202 - safety buffer, is this the same as headland?
line 378 - I don't understand this line
line 434 - what do you mean by ... and spike-1 weight..
line 439 misspelled waterlloged and the is to much space at ..from GS 30..
line 441 - what do you mean by (number of spikes-1..
line 445 - what do you mean by meter2
Reviewer 3 Report
It is nice paper about attempt to delineate N fertilization maps by using NDVI, especially the attempting to provide an answer for plots that do not show agreement between NDVI classified images and grain yield classified maps. However, it is some weakness for the conclusion that the mismatch may be the result of a delay in the onset of tillering, which causes a shortening of the tillering phase. Maybe a figure showing the comparison of the NDVI seasonal evolution in unrelated plots and in the related plots can give more clear evidence.
In detail:
English check should be more carefully for example
L16 a third should be one third and L18: another third should be another one-third.
L135: ANC fertilizer dose applied was 220 and 210 kg ha-1. These numbers are not coordinate with Table 1.
L173: annual rainfall should be monthly rainfall
L237: change “Geoeuskadi a repository for spatial data for the Basque Country” to geomorphological variables: Elevation, soil type and orthophoto
L264-266: Sentences are not well. Maybe it can be changed as “The degree of similarity between maps is quantified using KI, as it is considered a more reliable measure of agreement than simple percent agreement, since KI takes into account the likelihood of agreement occurring by chance.”
L384-404: Suggestion: give a figure to show the seasonal evolution of NDVI of unrelated plots and the related plots as Figure 5.
Round 2
Reviewer 2 Report
Dear Authors
Thank you for taking my comments into consideration for this new version. I only have a few comments:
delete vertical line in table 3
line 363: there is one number too much for the Babar reference
Author Response
We would like to thank the reviewer again for the effort made to review the document a second time. We have integrated the comment about the reference. However, we do not see any vertical line in table 3. I add a screenshot, where table 3 is shown.
